# Diagnostic performance of *Pf*HRP2/pLDH malaria rapid diagnostic tests in elimination setting, northwest Ethiopia

Melkamu Tiruneh Zeleke[1]*, Kassahun Alemu Gelaye[2], Adugna Abera Hirpa[3], Mahlet Belachew Teshome[3], Geremew Tasew Guma[3], Banchamlak Tegegne Abate[4], Muluken Azage Yenesew[1]

1 School of Public Health, Bahir Dar University, Bahir Dar, Ethiopia, 2 Institute of Public Health, University of Gondar, Gondar, Ethiopia, 3 Ethiopian Public Health Institute, Malaria, and Neglected Tropical Diseases Research Team, Addis Ababa, Ethiopia, 4 Medical Parasitology, Amhara Public Health Institute, Bahir Dar, Ethiopia

* mtiruneh089@gmail.com

**Data Availability Statement:** All relevant data are within the paper and its Supporting Information files.

## Abstract

Accurate diagnosis of malaria is vital for the effectiveness of parasite clearance interventions in elimination settings. Thus, evaluating the diagnostic performance of rapid diagnostic tests (RDTs) used in malaria parasite clearance interventions in elimination settings is essential. Therefore, this study aimed to evaluate the diagnostic performance of rapid diagnostic tests recently used in detecting malaria parasites in northwest Ethiopia. A facility-based cross-sectional study was conducted from November 2020 to February 2021 comparing *Pf*HRP2/pLDH CareStart malaria RDTs with light microscopy and polymerase chain reaction (PCR). Blood samples were collected from 310 febrile patients who attended the outpatient department and examined using CareStart RDTs, light microscopy, and PCR. Statistical analyses were performed using STATA/SE version 17.0. The sensitivity of *Pf*HRP2/pLDH CareStart malaria RDTs, regardless of species, was 81.0% [95% CI, 75.3, 86.7] and 75.8% [95% CI, 69.6, 82.0] compared to light microscopy and PCR, while the specificity was 96.8% [95% CI, 93.7, 99.9] and 93.2% [95% CI, 88.6, 97.8], respectively. The false-negative rate of CareStart malaria RDTs in comparison with light microscopy and PCR was 19.0% and 24.2%, respectively. The level of agreement beyond chance between tests was substantial, RDT versus microscopy was 75.0% and RDT versus PCR was 65.1%. The diagnostic performance of *Pf*HRP2/pLDH CareStart RDTs in detecting malaria parasites among febrile patients in the study area was below the recommended WHO standard. The limited diagnostic performance of RDTs in the malaria elimination area undoubtedly affects the impact of malaria parasite clearance interventions. Therefore, parasite clearance intervention like targeted mass drug administration with antimalarial drugs is recommended to back up the limited diagnostic performance of the RDT or replace the existing malaria RDTs with more sensitive, field-deployable, and affordable diagnostic tests.

**Funding:** This research received financial and material support from the Amhara Public Health Institute. The funding body had no role in the design of the study, data collection, data analysis, interpretation of results, and writing the manuscript. No author had received funds/awards, the Amhara Public Health Institute which is a government office had provided financial support through per diem payment for the training participants and during the data collection period for data collectors and supervisors. The payment was executed through the government payment policy and procedure. Laboratory supplies and reagents were supported by the Amhara Public Health.

**Competing interests:** The authors have declared that no competing interests exist.

## Introduction

Despite tremendous achievements that have been made in the fight against malaria, it remains a major public health problem in 85 countries and territories, including Ethiopia [1–3]. The World Health Organization (WHO) has set an ambitious goal of reducing malaria cases by 90% from the baseline in 2015 and eliminating it in 35 countries by the end of 2030 [4]. Ethiopia has also set a malaria elimination goal for a similar period [5].

Malaria control efforts have been focused on vector control strategies that reduce adult mosquito populations and human-mosquito contact and eradicate mosquito breeding habitats. Malaria elimination requires the clearance of the parasite that causes the disease from the human population, even though the mosquito population (the vector) may continue to be present [6, 7].

As malaria-endemic countries move toward elimination, there is a need for rapid and accurate diagnostic tools to detect malaria parasites in the human population so the parasites can be cleared with antimalarial drugs. Active monitoring and evaluation of the performance of diagnostic tools at the country level is necessary to guide policy on the selection and use of diagnostic tools in elimination settings [8, 9].

Malaria rapid diagnostic tests (RDTs) are the tools used to detect the malaria parasite. Parasite clearance interventions such as mass testing and treatment (MTAT) and focal testing and treatment (FTAT) are the active strategies that seek to clear malaria parasites from the human population based on appropriate stratification of malaria transmission intensity [10–13]. The effectiveness of the malaria parasite clearance interventions is highly dependent on the performance of the diagnostic tools [14].

Malaria RDTs are indirect tests based on the detection of antigens/enzymes produced by the malaria parasite such as *Plasmodium falciparum* histidine-rich protein 2 (*Pf*HRP2), which is specific to the *Plasmodium falciparum* species, and *Plasmodial* lactate dehydrogenase (pLDH) and *Plasmodial* aldolase, which are common among species other than *Plasmodium falciparum* [15, 16]. Malaria microscopy and polymerase chain reaction (PCR) are based on the detection of the parasite and genetic factor, respectively and they are considered the gold standard (references) diagnostic methods. Expert microscopists may detect ≥5–10 parasites per μl of blood and an average microscopist may detect only >100 parasites per μl. Malaria RDTs are expected to detect >100 parasites per μl whereas molecular tests (PCR) can detect 0.004–5 parasites per μl [17].

Documenting the diagnostic performance of malaria RDTs used for parasite clearance interventions in elimination settings is useful to inform decisions made by policymakers, program managers, and partners working for or with malaria elimination programs.

In a collaboration between the Amhara Regional State Health Bureau and PATH, a single round of MTAT followed by FTAT was implemented in six selected kebeles (the lowest administrative structure) of Amhara Regional State, Ethiopia. The selected intervention kebeles were Dehina Sositu and Yeginid Lomi in Bahir Dar Zuria District, Berhan Chora in Mecha District, Zengoba in Aneded District, Choresa in Kalu District, and Kumer Aftit in Metema District [10, 12].

CareStart RDTs, which can detect both *Pf*HRP2 and pLDH antigens in the human population, were used as a diagnostic tool for the interventions. However, evidence was limited on the diagnostic performance of CareStart RDTs and its effectiveness in elimination settings. Therefore, this study aimed to evaluate the performance and the effectiveness of CareStart RDTs in elimination settings and inform the decisions of policymakers, program managers, and partners working for or with the malaria elimination program.

## Material and methods

### Study area

The study was conducted in Andassa Health Center, Bahir Dar Zuria District, northwest Ethiopia from November 2020 to February 2021. Andassa Health Center was selected for close follow-up of data collection, and proper handling, transporting, and storage of samples to the regional reference laboratory. The health center is located at 11.5022611N and 37.4864854E with an altitude of 1,728 meters above sea level. The recorded daily temperature of the study area showed an average minimum temperature of 18.3˚C and an average maximum temperature of 28.4˚C. Malaria transmission is seasonal and unstable in the study area. *Anopheles gambiae* complex is the dominant species of mosquitoes that transmit malaria and *Plasmodium falciparum* and *Plasmodium vivax* are the two dominant parasite species in the study area [18, 19]. According to the district health office report, the health center and its six satellite health posts serve an estimated 60,490 population.

### Sample size determination

The sample size for the study was determined using a nomogram [20]., with a 5% level of significance. The estimated malaria prevalence was 18.3% and the sensitivity of CareStart RDTs was 93.2% based on the previous study conducted in the study area [21]. Therefore, the minimum representative sample size for the study was 310.

### Participants and eligibility

All consenting patients who attended the outpatient department and all suspected malaria cases were enrolled in the study. According to the national malaria guidelines, a suspected malaria case is a person with fever or a history of fever in the previous 48 hours or with a measured axillary body temperature of $\geq$ 37.5˚C if the patient is from a malaria-endemic area. If the patient is from a malaria-free area, fever, or history of fever in the previous 48 hours, or with a measured axillary body temperature of $\geq$ 37.5˚C and a history of travel to malaria-endemic areas within 30 days was considered as a suspected malaria case. All suspected cases were eligible for malaria parasitological tests and included in the study after obtaining consent/assent. A patient who had a history of recent treatment (one month preceding the survey) with antimalarial drugs was excluded from the study [22]. Only those eligible were informed about the study and enrolled after providing written consent/assent.

### Study procedures and laboratory analyses

**Specimen collection, processing, and testing.** Two milliliters of venous blood were collected from each study participant using ethylenediaminetetraacetic acid (EDTA) test tubes. A drop of blood (5 μl) was used to diagnose malaria using the CareStart malaria *Pf*HPR2/pLDH combo test kits. The tests were performed and interpreted following the manufacturer's instructions. The RDT is a qualitative immunochromatographic test that detects the antigens of both *Plasmodium falciparum* and *Plasmodium vivax*. The two species of *Plasmodium* account for nearly 100% of malaria in the study area.

Two separate drops of blood were placed on a frosted microscopic glass slide to prepare both thin and thick blood films for microscopy examination. The films were air-dried at room temperature and the thin film was fixed with absolute methanol. The thin and thick films were stained with 10% Giemsa solution for ten minutes and examined by experienced laboratory personnel using a light microscope. A slide was considered positive if at least one asexual blood stage of the *Plasmodium* parasite was identified. The presence of *Plasmodium* parasites

was ruled out if no parasites were observed after examining at least 100 microscopic fields with 100X objective lens. Two readings were conducted for each slide and discrepancies were resolved by the third reading by an independent trained and experienced microscopist.

Another two separate drops of blood were placed on Whatman 903 filter paper for PCR tests. The blood spots on the Whatman filter paper were air-dried and packed in a zip-lock bag with silica gel and stored in a freezer at -20˚C at the Amhara Public Health Institute until being transported to the Ethiopia Public Health Institute for PCR assay.

## DNA extraction

Genomic DNA extraction was performed using the Genesius™ Micro gDNA Extraction Kit (Geneaid Biotech Ltd.) from a dried blood spot (DBS) sample. A 6 mm diameter circle was punched out from a DBS with a single-hole paper puncher and then transfer to a 1.5 ml micro-centrifuge tube for processing, as per manufacturer instructions, and DNA was eluted with a 50 μl volume of elution buffer and stored at -20˚C until analyzed.

**DNA amplification with multiplex real-time PCR.** The PCR amplification was done by using primer and probes that target genes coding for the small subunit of ribosomal RNA specific for *Plasmodium* genus (18S rRNA) and *P. vivax* (P.v18S rRNA), and the *var* gene acidic terminal sequence (*var*ATS) specific for *P. falciparum*. The human RNaseP sequence was targeted as an internal control to assess the quality of DNA extraction and qPCR amplification. TaqMan fluorescence-based DNA amplification and detection were performed using the QuantStudio 5 Real-time PCR system (ThermFisher Scientific).

The real-time PCR assay was run in two rounds; during the first run, all samples were tested by using pan-*Plasmodium*-specific primers for the detection of the *Plasmodium* genus, and the second multiplex real-time PCR run was done for the detection of *Plasmodium* species using *P. falciparum*- and *P. vivax*-specific primers. Briefly, each reaction mixture was prepared by mixing 2 μl of purified DNA template, 5 μl Luna Universal Probe qPCR Master Mix (New England Biolabs, Inc.), 2 μl PlasQ Primer mix, and 1 μl molecular biology grade water with a final reaction mixture volume of 10 μl. The PCR amplifications were carried out using thermal cycling conditions. The first PCR run was 95˚C for 1 minute, followed by 45 cycles of 95˚C for 15 seconds and 57˚C for 45 seconds, and the second run was 95˚C for 1 minute, followed by 45 cycles of 95˚C for 15 seconds and 53˚C for 45 seconds.

The 3D7 DNA standard was run in each experiment and used as a positive control and nuclease free water was used as a negative control. For PCR runs, the positive control had to have a Ct (cycle of threshold) value of 25 to 30, and all samples having a Ct value of < 30.0 for HsRNaseP qualified as a run. Samples with a Ct value between 12 to 40 and a sigmoidal shape amplification curve were considered positive.

Parasite density was calculated for *Plasmodium falciparum* based on a protocol developed in-house. The lyophilized material was dissolved in 0.5 ml sterile nuclease-free water according to National Institute for Biological Standards and Control protocol. After reconstitution, the standard concentration was equivalent to a parasite density of $5 \times 10^5$ parasites per μL. A serial dilution was prepared in nuclease-free water. A qPCR assay was performed on triplicates for each standard concentration and Ct values were used to prepare a standard curve.

**Quality control and assessment.** Laboratory investigations were carried out at Andassa Health Center, Amhara Public Health Institute, and Ethiopia Public Health Institute. Malaria rapid diagnostic testing was performed at the health center according to the recommended standard. The microscopic examinations were done at the health center and the Amhara Public Health Institute. Blinding was imposed on the laboratory technicians and discrepancies were resolved using a third reader. In general, quality control was done for all laboratory

**Table 1. A 2 X 2 table to evaluate the performance of diagnostic tests.**

| CareStart RDT | Gold standard test (microscopy or PCR) | | Total |
|:---:|:---:|:---:|:---:|
| | Positive | Negative | |
| Positive | a (TP) | b (FP) | a+b |
| Negative | c (FN) | d (TN) | c+d |
| Total | a+c | b+d | a+b+c+d |

methods and procedures according to the standard operating procedures and manufacturer's instructions.

**Data management and statistical analysis.** The data were collected using SAMSUNG S3 smart mobile phone equipped with a questionnaire developed using ODK Collect v1.25.1 and the collected data were downloaded from a locally created server. Statistical analysis was performed using STATA/SE version 17.0. The performance of the diagnostic tests was evaluated using sensitivity, specificity, positive predictive value, and negative predictive value. Cohen's Kappa was calculated to measure the agreement between tests beyond chance.

The CareStart RDT results and the gold standard tests were categorized as true positive (TP), true negative (TN), false positive (FP), or false negative (FN). Sensitivity was calculated as TP/ (TP + FN) and specificity as TN/ (TN + FP). Positive predictive value was defined as TP/ (TP + FP) and negative predictive value as TN/ (TN + FN). False-positive rate was calculated as FP/ (FP + TN) and false-negative rate was calculated as 1- (TN/ [TN + FP]). Test accuracy, the proportion of all tests that yielded a correct result, was calculated as (TP + TN) / (TP + TN + FP + FN) (Table 1) [14].

## Ethical considerations

Ethical approval was obtained from the Institutional Review Board (IRB) of the College of Medicine and Health Sciences, Bahir Dar University, with protocol number 00223/2020. A support letter was written to the study district and health center. Written informed consent was obtained from each study participant. Assent was obtained for the age group between 12 to 17 after obtaining consent from their parents and guardian. Malaria-positive cases were treated according to the national treatment guidelines at the health center. No unique identifier was attached to the individual data and unique identifiers were kept confidential.

## Results

A total of 310 patients attended the outpatient department of the health center and those who fulfilled the inclusion criteria were screened. Seven samples were discarded due to poor quality. Approximately 58.4% of the participants were male and their mean age was 24.8 (SD±12.5) years. All the study participants were rural residents (Table 2).

Of a total of 303 blood samples, 194 (64.0%) tested positive for *Plasmodium* infection by at least one of the diagnostic methods. CareStart malaria RDT test results revealed that 83 (27.4%), 63 (20.8%), and 3 (1.0%) participants were positive for *Plasmodium falciparum*, *Plasmodium vivax*, and mixed infections, respectively. Microscopy tests detected *Plasmodium falciparum*, *Plasmodium vivax*, and mixed infections in 123 (40.6%), 52 (17.2%), and 4 (1.3%) participants, respectively. The molecular test (PCR) detected *Plasmodium falciparum* in 129 (42.6%), *Plasmodium vivax* in 39 (12.9%), and mixed infections in 18 (5.9%) participants (Table 3 and Fig 1).

Among the total samples, 149 positive cases were detected by malaria RDT, of which 145 were found positive by microscopy with a difference in species identification. Additional

**Table 2. Sociodemographic characteristics of the study participants at Andassa Health Center, Bahir Dar Zuria district, northwest Ethiopia from November 2020 to February 2021.**

| Characteristics (N = 310) | Category | N (%) |
|---|---|---|
| Sex | Male | 177 (58.4) |
| | Female | 126 (41.6) |
| Age (Year) | < 5 | 24 (7.9) |
| | 5–14 | 54 (17.8) |
| | 15–24 | 101 (33.3) |
| | 25–34 | 70 (23) |
| | 35–44 | 36 (11.9) |
| | 45–67 | 18 (5.9) |
| Residence | Rural | 303 (100) |
| | Urban | 0 (0) |

positive cases (34) were identified by microscopy and PCR with a slight difference in species identification between the microscopy and PCR. From those samples identified as negative by RDT and microscopy, 11 samples were found positive by PCR diagnostic test (Fig 1).

## Performance of PfHRP2/pLDH CareStart RDTs

Compared to microscopy, the sensitivity and specificity of the RDTs were 81.0% and 96.8% respectively. The positive and negative predictive values were 97.3% and 77.9%, respectively. The sensitivity and specificity of RDT were 75.8% and 93.2%, respectively as compared to the molecular test (PCR). The positive and negative predictive values were 94.6% and 70.8%, respectively, as compared to the PCR tests. The test accuracy rate of the RDTs as compared to microscopy and PCR was 87.5% and 82.5%, respectively. The level of agreement beyond chance between RDT and microscopy, and RDT and PCR, was 75.0% and 65.1%, respectively (Table 4).

## Discussion

The findings provided evidence of the limited diagnostic performance of RDTs in malaria elimination settings, which then places a clear limitation on the effectiveness of parasite clearance interventions.

Microscopy remains the gold standard diagnostic test for malaria in clinical settings in elimination areas as compared to the molecular test (PCR). The study found that the sensitivity of PfHRP2/pLDH CareStart RDTs, in comparison with microscopy and PCR, was below the recommended standard by WHO. The recommended sensitivity is $\geq$ 95% at $\geq$ 100 parasites per μl for Plasmodium falciparum [15, 16, 23]. In this study, the use of CareStart RDTs resulted

**Table 3. Test positivity rate by different diagnostic techniques at Andassa Health Center, Bahir Dar Zuria District, northwest Ethiopia from November 2020 to February 2021.**

| Results | Diagnostic methods | | | Matched result |
|---|---|---|---|---|
| | CareStart RDT | Microscopy | PCR | |
| P. falciparum | 83 (27.4%) | 123 (40.6%) | 129 (42.6%) | 74 (24.4%) |
| P. vivax | 63 (20.8%) | 52 (17.2%) | 39 (12.9%) | 35 (11.6%) |
| Mixed | 3 (1.0%) | 4 (1.3%) | 18 (5.9%) | 3 (1.0%) |
| Total Positive | 149 (49.2%) | 179 (59.1%) | 186 (61.4%) | 112 (40.0%) |
| Total Negative | 154 (50.8%) | 124 (40.9%) | 117 (38.6%) | 109 (36.0%) |

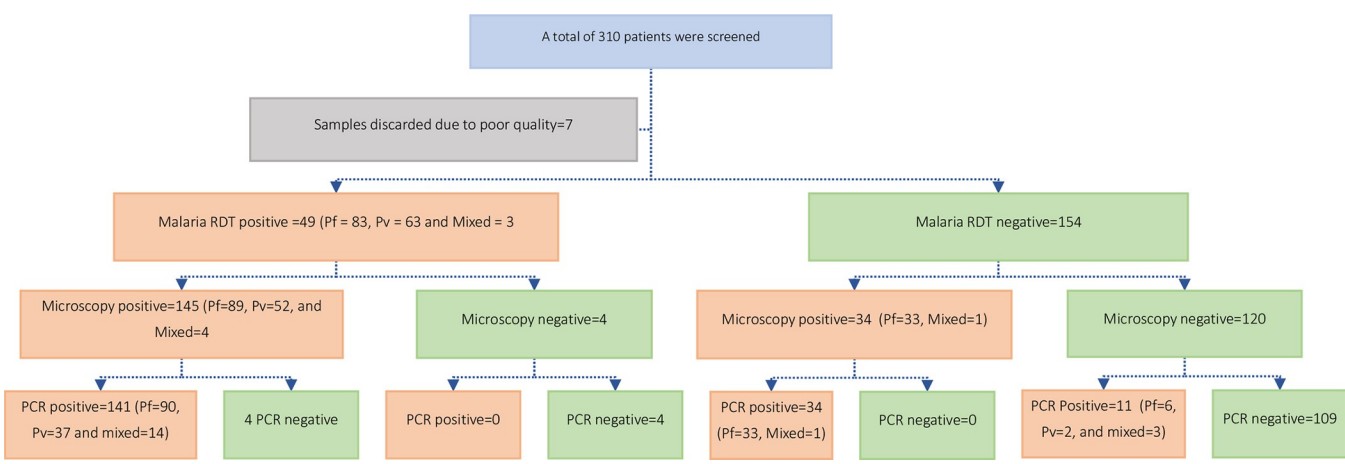

**Fig 1. A study flowchart that showed patients' enrollment, CareStart RDTs, Microscopy, and PCR test results.**

in the under-diagnosis of malaria infection in 19.0% and 24.2% of patients as compared to microscopy and PCR, respectively. The low sensitivity of the RDT might have a serious impact on the effectiveness of malaria mass testing and treatment followed by focal testing and treatment interventions. This finding is comparable with previous studies [24–26]. and contradicts other previous studies [21, 27–30]. The possible reasons for the discrepancy could be due to a difference in the prevalence of malaria, type of RDTs, transport and storage of RDTs, sample handling, and individual reading difference.

The sensitivity of a diagnostic test is affected by the rate of false negatives [31]. In this study, CareStart RDTs had a false negative rate of 24.2% and failed to detect *Plasmodium falciparum* species in 61 (49.6%) compared to the molecular test (PCR). This could be because of the deletion of the HPR2 gene, which has affected malaria elimination efforts in the country. The findings of this study are consistent with recent studies conducted elsewhere, including in Ethiopia [32–34].

The sensitivity of RDTs is affected by the parasite densities in blood circulation [17]. In this study, *Plasmodium falciparum* parasite detection performance of *Pf*HRP2/pLDH CareStart RDTs was below (58%) the WHO standard ($\geq$95% at $\geq$ 100 parasites/μl of blood) though it was able to detect the parasite in below 100 parasites (Table 5) [16, 35].

The specificity of CareStart RDTs in comparison with microscopy and PCR was above the recommended standard by the WHO. The finding is comparable with previous studies conducted elsewhere [21, 36]. but in contrast with the previous study done in Ghana [28].

**Table 4. The performance of *Pf*HRP2/pLDH CareStart RDTs as compared to microscopy and PCR as a gold standard.**

| CareStart RDTs | Microscopy | PCR |
|---|---|---|
| Sensitivity, % (95% CI) | 81.0 (75.3, 86.7) | 75.8 (69.6, 82.0) |
| Specificity, % (95% CI) | 96.8 (93.7, 99.9) | 93.2 (88.6, 97.8) |
| Positive predictive value, % | 97.3 (94.7, 99.0) | 94.6 (91.0, 98.2) |
| Negative predictive value, % | 77.9 (71.3, 84.5) | 70.8 (63.6, 77.9) |
| Kappa, % | 75.0 (70.1, 79.9) | 65.1(59.7, 70.5) |
| False-negative rate, % | 19.0 | 24.2 |
| False-positive rate, % | 3.2 | 6.8 |
| Accuracy, % | 87.5 | 82.5 |

**Table 5. The performance of *Pf*HRP2/pLDH CareStart RDTs in detecting *Plasmodium falciparum* as compared to the parasite density in the blood.**

| Parasite/µl of blood | PCR positive for *Pf* | RDTs positive for *Pf* | Accuracy (%) |
|---|---|---|---|
| < 100 | 59 | 25 | 42.4 |
| 100–500 | 45 | 28 | 62.2 |
| 501–1000 | 21 | 11 | 52.4 |
| 1001–4000 | 14 | 7 | 50.0 |
| ≥ 4000 | 8 | 5 | 62.5 |
| Total | 147 | 76 | 52.0 |

The specificity of a diagnostic test is affected by the rate of false positives. In this study, RDTs had a false positive rate of 3.2% and 6.8% as compared to microscopy and PCR, respectively. The potential reasons for false positivity include cross-reaction with rheumatoid factors and incomplete and recent treatment of malaria [22, 37–39]. We tried to exclude those patients who had a history of incomplete and recent treatment with antimalarial drugs; therefore, the false positive rate could be because of cross-reaction with rheumatoid factors.

## Conclusion

The diagnostic performance of *Pf*HRP2/pLDH CareStart RDTs in detecting malaria parasites among febrile patients in elimination settings was found below the recommended WHO standard. Further studies are essential at the community level in the malaria-elimination areas. The limited diagnostic performance of RDTs in malaria elimination settings undoubtedly affects the impact of malaria parasite clearance interventions (MTAT and FTAT). Therefore, parasite clearance interventions like targeted mass drug administration with antimalarial drugs is recommended to back up the limited diagnostic performance of the RDT or replace the existing malaria RDTs with more sensitive, field-deployable, and affordable diagnostic tests.

## Supporting information

**S1 Data.**
(XLSX)

## Acknowledgments

We would like to thank the Amhara Public Health Institute for partially funding the project and for the laboratory support provided. We also thank the Ethiopian Public Health Institute for supporting the molecular tests. We gratefully acknowledge all study participants and data collectors. Finally, we would like to extend our heartfelt acknowledgment to Belendia Abdissa for his unreserved IT support.

## Author Contributions

**Conceptualization:** Melkamu Tiruneh Zeleke, Kassahun Alemu Gelaye, Muluken Azage Yenesew.

**Data curation:** Melkamu Tiruneh Zeleke.

**Formal analysis:** Melkamu Tiruneh Zeleke.

**Funding acquisition:** Melkamu Tiruneh Zeleke.

**Investigation:** Melkamu Tiruneh Zeleke.

**Methodology:** Melkamu Tiruneh Zeleke, Kassahun Alemu Gelaye, Adugna Abera Hirpa, Mahlet Belachew Teshome, Geremew Tasew Guma, Banchamlak Tegegne Abate, Muluken Azage Yenesew.

**Software:** Melkamu Tiruneh Zeleke.

**Supervision:** Melkamu Tiruneh Zeleke.

**Validation:** Melkamu Tiruneh Zeleke.

**Writing – original draft:** Melkamu Tiruneh Zeleke.

**Writing – review & editing:** Melkamu Tiruneh Zeleke, Kassahun Alemu Gelaye, Adugna Abera Hirpa, Mahlet Belachew Teshome, Geremew Tasew Guma, Banchamlak Tegegne Abate, Muluken Azage Yenesew.

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
