## [Decision Letter · Decision Letter 0]

20 Mar 2023

PGPH-D-22-01932

Diagnostic performance of PfHRP2/pLDH CareStart malaria rapid diagnostic tests in elimination setting, northwest Ethiopia

Dear Dr. Zeleke

Thank you for submitting your manuscript to PLOS Global Public Health. After careful consideration, we feel that it has merit but does not fully meet PLOS Global Public Health’s publication criteria as it currently stands. Therefore, we invite you to submit a revised version of the manuscript that addresses the points raised during the review process.

We look forward to receiving your revised manuscript.

Kind regards,

Om Prakash Singh, PhD

Academic Editor

Journal Requirements:

1. Please indicate the full and correct funding information for your study and confirm the order in which funding contributions should appear.

Additional Editor Comments (if provided):

Reviewers' comments:

Reviewer's Responses to Questions

**Comments to the Author**

1. Does this manuscript meet PLOS Global Public Health’s publication criteria? Is the manuscript technically sound, and do the data support the conclusions? The manuscript must describe methodologically and ethically rigorous research with conclusions that are appropriately drawn based on the data presented.

Reviewer #1: Yes

Reviewer #2: Partly

2. Has the statistical analysis been performed appropriately and rigorously?

Reviewer #1: No

Reviewer #2: Yes

3. Have the authors made all data underlying the findings in their manuscript fully available (please refer to the Data Availability Statement at the start of the manuscript PDF file)?

Reviewer #1: Yes

Reviewer #2: No

4. Is the manuscript presented in an intelligible fashion and written in standard English?

Reviewer #1: Yes

Reviewer #2: Yes

5. Review Comments to the Author

Reviewer #1: Thank you for the opportunity to review this paper. Here are some comments for authors' consideration, hopefully to further strengthen the paper.

Major comments:

1. CareStart is a trademark protected by applicable laws. Please strongly justify the use of trademarked names, instead of just saying PfHRP2/pLDH, in your manuscript, otherwise please omit them.

2. Please elaborate what is known about malaria transmission intensity, parasite and vector diversity in the catchment area and population served by Andassa Health Center (AHC). This critical for a better understanding of the diagnostic performance of PfHRP2/pLDH.

3. A lot more descriptive data needs to be shown before the diagnostic performance results can be credibly interpreted. Please add a table first to describe your study population ( children vs. adults) and study site ( non-malaria areas vs malarious areas) and stratify the diagnostic performance findings across these variables.

4. The authors state AHC was purposively selected, so please justify the use of inferential statistics in your analysis? If your sample was purposively selected the 95% confidence interval and p-values become uninterpretable. Please justify or revise accordingly.

Minor comments:

1. The term “rate” should not be used to describe proportions or percentages. In descriptive epidemiology the term “rate” implies person-time, which is not the case in this manuscript. Please appropriately revise.

2. Please define the term ‘elimination settings”.

3. Line 99: what does “als” mean. One should not have to guess what it means.

4. Please support suspected malaria case definition with relevant references.

5. Please clearly state how consent was sought from under-age population (children).

6. Line 126: the test does not detect plasmodium it detects certain antigens of the plasmodium. Please rephrase for clarity and accuracy.

7. Laboratory methods: Please support all your laboratory methods (RDTs, microscopy and PCR) with references to reflect that standardized protocols were followed. Any deviations from standardized protocols should be clearly stated.

8. Formulas for diagnostic accuracy should be supported with relevant references.

9. Fig1 and methods section should also clearly state and show how many suspected malaria cases were screened and how many were excluded for receiving antimalarials within 4 weeks before arriving at the sample size of 310.

10. Table 4. Please clearly justify class interval used in the frequency table. Parasite density cutoffs and class intervals needs to reflect underlying biological, pathological, or operational basis.

Reviewer #2: 1. In discussing study area, line 97 describes Andassa health center as a representative of all transmission settings. Please clarify this.

2. Please include a picture of the nomogram as a figure in the publication

3. In the methods discussing parasite density calculation, lines 169-174 does not include information on how the standard curve generated via PCR was used to arrive at contents of table 4

4. Table 4 should be part of results and not discussion.

5. Specific mention should be made about the benefits and opportunities for use of Malaria RDTs in your discussions and conclusion to avoid giving the impression that they are diagnostically useless

6. PLOS authors have the option to publish the peer review history of their article (what does this mean?). If published, this will include your full peer review and any attached files.

**Do you want your identity to be public for this peer review?** For information about this choice, including consent withdrawal, please see our Privacy Policy.

Reviewer #1: No

Reviewer #2: **Yes: **Anthony Agbakizu Ahumibe

---

## [Editor Report · Decision Letter 1]

24 May 2023

PGPH-D-22-01932R1

Diagnostic performance of PfHRP2/pLDH malaria rapid diagnostic tests in elimination setting, northwest Ethiopia.

Dear Dr. Zeleke 

Thank you for submitting your manuscript to PLOS Global Public Health. After careful consideration, we feel that it has merit but does not fully meet PLOS Global Public Health’s publication criteria as it currently stands. Therefore, we invite you to submit a revised version of the manuscript that addresses the points raised during the review process.

We look forward to receiving your revised manuscript.

Kind regards,

Om Prakash Singh, PhD

Academic Editor

Journal Requirements:

Additional Editor Comments (if provided):

***Please revise the article as this revised version of draft has errors. For examples- Line 199: references is still missing, table 4 is still in discussion section etc.
---

## [Editor Report · Decision Letter 2]

8 Jun 2023

Diagnostic performance of PfHRP2/pLDH malaria rapid diagnostic tests in elimination setting, northwest Ethiopia.

PGPH-D-22-01932R2

Dear Dr. Zeleke,

We are pleased to inform you that your manuscript 'Diagnostic performance of PfHRP2/pLDH malaria rapid diagnostic tests in elimination setting, northwest Ethiopia.' has been provisionally accepted for publication in PLOS Global Public Health.

Best regards,

Om Prakash Singh, PhD

Academic Editor